# Metabolism and Immune Suppressive Response in Liver Cancer

**DOI:** 10.3390/biomedicines13061461

**Published:** 2025-06-13

**Authors:** Patrizio Caini, Vinicio Carloni

**Affiliations:** Department of Experimental and Clinical Medicine, University of Florence, 50121 Florence, Italy

**Keywords:** glycolysis, hypoxia, macrophages, regulatory T cells, epigenetics

## Abstract

Hepatocellular carcinoma (HCC) constitutes more than 90% of the primary tumor of the liver. Metabolic reprogramming is decisive in promoting HCC development. The new metabolic program drives the surrounding immune cells to an immune suppressive commitment, enabling tumor survival. The enhanced metabolic activity of cancer cells leads to competition for essential nutrients, depriving non-malignant cells of critical resources. Simultaneously, the accumulation of metabolic byproducts within the tumor microenvironment (TME) selectively favors innate immune responses while impairing adaptive immunity. Recent advances in cancer immunotherapy underscore the importance of targeting both immune cell function and metabolic pathways. In this context, reprogramming the metabolism of effector and regulatory immune cells represents a promising therapeutic avenue. This review focuses on a relatively underexplored aspect of liver cancer immunology, the immunosuppressive role of tumor-associated macrophages (TAMs) and regulatory T cells (Tregs) driven by metabolic alterations and how these mechanisms contribute to the suppression of effective anti-tumor immune responses.

## 1. Introduction

Hepatocellular carcinoma (HCC) represents one of the deadliest cancers worldwide [1]. Curative therapy with tumor ablation, resection, or liver transplantation is dependent on early detection. Recent data suggest that malignant transformation in HCC is not primarily caused by cancer driver mutations, but also epigenetic factors account for the transitions from cirrhotic tissues, dysplastic nodules, and HCC [2,3]. Additionally, in metabolic-dysfunction-associated fatty liver disease (MAFLD), a relevant public health problem in western countries, there is increasing evidence that HCC arises in non-cirrhotic livers and the pathogenetic mechanisms are scarcely defined in this population [1]. Notably, in these patients, an ineffective/detrimental action of immunotherapy with anti-PD-1/PD-L1 immune checkpoint inhibitors has also been described [4]. All together, these observations reflect the challenge to define exactly the relationships between liver cancer and its environment and counteract the immune suppressive nature of TME.

## 2. Tumor Microenvironment Components

The TME is highly heterogeneous and compositionally different among HCC grades. The major components of TME are blood vessels and various “accessory” non-neoplastic cells such as immune cells, fibroblasts, and stem cells. TME is also enriched by a variety of signaling molecules such as chemokines, cytokines, and enzymes; together, they play a pivotal role during cancer progression and metastasis. The cellular component is embedded in the extracellular matrix (ECM); therefore, cancer cells, accessory cells, and elements of ECM all together collaborate in tumor progression. ECM consists of distinct types of collagen and non-collagenous glycoproteins such as elastin, laminins, and fibronectin. Importantly, ECM proteins store growth factors such as HGF, TGF-β, EGF, and VEGF. ECM turnover results in the release and activation of growth factor signaling cascades, promoting epithelial proliferation and, most likely, cancer development in chronically injured tissues, as observed in the genesis of HCC (Figure 1).

## 3. HCC Prepares the Ground: Acidity and Hypoxia

### 3.1. Acidity

Normal healthy cells gain energy through oxidative phosphorylation (OXPHOS), and glycolysis is repressed under normoxia. In contrast, cancer cells tend to switch their metabolism into glycolysis. The first concept of such metabolic reprogramming in cancer cells dates backs to the Otto Warburg observations, indicating that most cancer cells produce their energy mainly through “aerobic glycolysis”, even in the presence of abundant oxygen [5]. Currently, we can consider this aspect correlative; generally, cancer cells require a large quantity of energy to support their multifaced metabolism, and glucose with its high concentration in the blood turns out to be a predominant source of energy. HCC cells increase the expression of glycolytic enzymes and glucose transporters to facilitate the uptake of extracellular glucose [6]. In brief, oxygen-deprived or well-oxygenated tumors have prevalent glycolysis as the primary source of ATP production rather than the expected OXPHOS. After glucose, glutamine is the most abundant amino acid in the blood. The metabolic importance of glutamine is emphasized by the fact that glutamine metabolism is required also as a component of a signaling process to speed up proliferation [7]. HCC also requires glutamine in addition to glucose for proliferation and survival. Glutamine, after being converted to glutamate and its subsequent conversion into α-ketoglutarate, can enter the TCA pathway to yield ATP. Moreover, α-ketoglutarate is associated with epigenetic regulation of gene expression in cancer cells as deficiency in glutamine can trigger DNA demethylation, as observed in melanoma [8]. Glutamine plays biosynthetic roles such as the production of amino acids, nucleotides, and fatty acids, influencing redox homeostasis and the hexosamine pathway [9,10]. Glycolysis results in the build-up of lactate, which can be used as fuel by HCC [10] or by nearby cells surrounding the tumor. A high metabolic rate of cancer cells, as well as easy diffusion due to an inadequate blood vessel structure, leads to a high concentration of H^+^ in TME. Tumor acidosis is a key player in the somatic evolution of transformed tissue and progression to malignancy [11,12,13]. The pH of extracellular fluid surrounding the cancer mass is between 6.3 and 7.0 in comparison to the physiological pH of 7.35–7.45 around the non-cancerous tissues, which is necessary for normal function of the cells. As a consequence of tumor acidity, the TME promotes the selection of cells that are able to survive in this hostile environment to the detriment of the other cell types. On the other hand, the high metabolic demand of cancer cells by using glycolysis leads to a build-up of H^+^ ions in the TME and represents a significant advantage to cancer cells to the extent that the inhibition of glycolysis has been proposed as a therapeutic possibility for cancer treatment [13,14,15].

### 3.2. Hypoxia

In TME, the oxygen concentration is significantly lower than normal healthy tissues, and the median oxygen pressure often decreases to 0–20 mmHg, while in normal tissues, the oxygen pressure is about 40 mmHg and nearly 100 mmHg (~13%) in arterial blood. As such, a hypoxic zone is principally formed due to the unrestrained proliferation of cancer cells and inadequate neo-angiogenesis [16,17]. The hypoxia-inducible transcription factor (HIF) signaling often preserves cells from differentiation and promotes angiogenesis [18]. Moreover, hypoxia can induce *HIF*-related genes of stemness and pluripotency such as *MYC*, *OCT4*, *SOX2*, and *NANOG* [16]. Lastly, hypoxia forces cells towards genetic instability, speeding up the accumulation of mutations that can eventually result in drug resistance [17]. Under normoxia, HIF-1 is degraded by von Hippel–Lindau (VHL) E3 ubiquitin ligase; in this regard, HIF-1 modulation is considered a promising strategy in cancer treatment [19,20]. In the hypoxic regions of cancer, cells divide more slowly in comparison to the well oxygen-rich regions; hence, hypoxic cancer cells show greater resistance to chemotherapies and radiation, as is the case of HCC patients subjected to TACE (trans-arterial chemoembolization) [21,22]. The same mechanism restrains the functionality of effector T cells when prolyl-hydroxylase proteins sense hypoxia and subsequently downregulates glycolysis, which prevents the CD8^+^ T cells and CD4^+^ cells from operating effectively [23]. Hypoxia downregulates the glycolysis genes of immune cells and, at the same time, induces the secretion of lactate by neoplastic cells into TME. Lactate inhibits the activation of CD8^+^ cytotoxic T cells and NK cells by preventing the recycling of glycolytic by-products in these cells, all in a vicious circle [24]. Furthermore, due the heterogenous metabolic aspects of the tumor, highly glycolytic tumor regions parallel regions of hypoxia, contributing to the acidic TME [25,26]. This aspect can be ascribed to lactate excretion, which depends on proton co-transporters; and the MCT family of transporters, consisting of four members (*MCT1-4*) of the *SLC16* gene family, has been shown to link H^+^ transport to lactate via electro-neutral H^+^/lactate transporters. MCT-1 and MCT-4 represent the most relevant players in HCC. Although MCT-1 is primarily used for lactate import or export in most tissues, MCT4 is the dominant isoform found in cancers [27,28,29].

## 4. Immune Surveillance and Metabolism

Glucose, fatty acids, and glutamine considered as nutrients represent also key components of immune cell metabolism [30,31,32]. Immune cells consume these nutrients to generate ATP and necessary building blocks for proteins, nucleic acids, and lipids (Figure 2 and Table 1). In general, immune T cells are subdivided into naïve, effector, memory, and regulatory T cells; each of them shows a distinguished profile of metabolite requirements and uses different metabolic pathways. As an example, inactivated (naïve) T cells require adenosine triphosphate (ATP) and a small number of metabolites to maintain their basic physiological function and actin cytoskeleton rearrangement for migration through the body [31]. Therefore, naïve T cells employ the OXPHOS pathway to fully oxidize glycolysis products. At this phase, glucose metabolism and glutamine oxidation account for 25% and 75% of the total production of lactate, respectively [32,33,34,35]. Since the naïve T cells are activated by appropriate antigen, the resulting effector T cells have a significantly higher metabolism activity compared to naïve T cells [36,37,38]. In addition to consuming larger amounts of glucose and glutamine, effector T cells now require further amino acids such as serine, arginine, and leucine [39,40,41,42]. Moreover, the effector cells start distancing from OXPHOS as their main source of energy; now, glucose metabolism and glutamine oxidation account for 67% and 33% of the lactate produced, respectively [43]. The rate of glucose metabolism increases to enhance glycolysis and, at the same time, stimulates the pentose phosphate pathway to provide NADPH for lipid synthesis and nucleic acid intermediates; all these changes enable the effector T cells to proliferate upon antigen activation [44,45]. After the successful removal of the antigen, a small number of effector T cells differentiate into memory T cells [46,47,48]. This procedure requires metabolic reprogramming that is characterized by enhanced oxidation of fatty acid by mitochondria. In contrast to effector T cells that need OXPHOS to be able to effectively engage in cell proliferation [46], memory T cells rely on OXPHOS to meet metabolic demands [47], but they use fatty acids to promote this process [49]. Fatty acid oxidation (FAO) represents a metabolic advantage for memory T cell survival and rapid response after encountering the antigen [49,50]. Interestingly, these cells do not acquire extracellular free fatty acids, they consume extracellular glucose to fuel FAO and OXPHOS, meaning that these cells synthesize fatty acid that is needed for mitochondrial FAO [51]. This is further supported by the finding that the lysosomal acid lipase that hydrolyzes cholesterol esters and triglyceride to produce fatty acids and cholesterol free in the cell is expressed in CD8^+^ memory T cells [51]. These latter cells also benefit from glycogenesis, which allows them to convert glucose to glycogen [52]. The resultant glycogen goes through glycogenolysis to generate glucose-6-phosphate; next, the pentose phosphate pathway generates abundant NADPH for fatty acid synthesis, ensuring the creation and maintenance of memory T cells [53,54,55,56].

## 5. Nutrients and Cell Signaling Pathways

The influence of nutrients is not limited to metabolism per se as it can also impact various nutrient-sensitive signaling pathways [57,58,59]. The 5′AMP-activated protein kinase (AMPK) is a signaling pathway that responds to energetic and/or cellular stress [60]. Interestingly, AMPK senses the concentration of glucose and fructose-1,6-bisphosphate independently of changes in adenine nucleotides [61]. Glucose deprivation can rapidly activate AMPK in both helper and cytotoxic T cells to modulate T-cell metabolism and effector function [62]. Similarly, low glutamine levels can stimulate AMPK, which restrains the activity of the mammalian Target of Rapamycin Complex 1 (mTORC1) [63]. Under energy stress, AMPK restrains the activity of mTORC1 in T cells and restricts the production of IFN-γ in both helper and cytotoxic T cells [64]. mTORC1 is a phosphatidylinositol kinase-related kinase (PIKK) complex that plays a cardinal role in shaping and regulating the functions of both innate and adaptive immune cells; part of these duties includes controlling the differentiation and survival of T cells, dictating the outcome of T regulatory cells (Treg), and acting in the proliferation, maturation, and properties of NK cells [65]. The concentration of several amino acids, including leucine, arginine, and glutamine regulating the complex mTORC1 is able to handle the de novo synthesis of serine and glycine [66,67,68]. Ultimately, glucose deprivation in TME inhibits the inflammatory function of both cytotoxic and memory T cells as well as NK cells while promoting the differentiation of Treg; all together, this promotes an immunosuppressive microenvironment. Besides fructose-1,6-bisphosphate, glycolysis-related phosphoenolpyruvate (PEP) can regulate a signaling pathway that contains both Ca^2+^ and a downstream transcription factor known as nuclear factor of activated T cells (NFAT). PEP represses the activity of sarco/ER Ca^2+^-ATPase (SERCA), which normally transfers Ca^2+^ from the cytosol of the cell to the sarcoplasmic reticulum. The presence of Ca^2+^ in the cytoplasm prolongs the duration of Ca^2+^-NFAT signaling. However, in a glucose-poor microenvironment, a lower concentration of PEP would cause an increase in the activity of SERCA, which maximizes the transport of Ca^2+^ to sarcoplasmic reticulum. A defective Ca^2+^-NFAT signaling would hinder naive T cells from exhibiting sufficient effector properties. Furthermore, amino acids and glucose also regulate protein O-GlcNAcylation. They act as precursors of uridine diphosphate *N*-acetylglucosamine (UDP-GlcNAc), which is a substrate for cellular O-GlcNAc glycosyltransferase, which regulates post-translational modification. O-GlcNAcylation plays a central role in the key stages of the development and activation of T cells. Some of the signaling molecules that are affected by the dysregulation of O-GlcNAcylation include c-Myc, NF-κB, and NFAT [37,69]. One-carbon (1C) metabolism integrates nutrients by the redistribution of carbon groups from substrates such as serine and glycine to folate metabolites and generates different cellular components such as glutathione (GSH) and *S*-adenosyl-L-methionine (SAMe). These substrates maintain cellular redox states and are critical for DNA and RNA biosynthesis [70]. Serine is either obtained as nutrient or endogenously generated by de novo synthesis. De novo synthesis starts with the conversion of 3-phosphoglycerate (3-PG), which is an intermediate of glycolysis, to 3-phosphohydroxypyruvate (3-PHP) by phosphoglycerate dehydrogenase (PHGDH) [68]. Cell proliferation is accompanied by increases in 3-PG, and the overexpression of the PHGDH enzyme occurs in cancer cells including HCC [6,71,72,73]. The lack of serine and glycine as nutrients promotes endogenous serine and glycine metabolism and causes a decrease in reduced GSH and an increase in reactive oxygen species (ROS) levels. In fact, serine depletion promotes O_2_ consumption and OXPHOS in mitochondria [74]. Serine is converted to glycine by two isoforms of serine hydroxymethyl-transferase, SHMT1 and SHMT2, which correspond to the cytosolic and mitochondrial forms of the enzyme, respectively [75,76,77]. Glycine is used as a metabolite for purine production and GSH synthesis [78]. The folate and methionine cycles mediate redistribution of 1C/methyl groups, which are critical for the biosynthesis of purines, pyrimidines, GSH, NADPH, and SAMe. One-carbon metabolism also provides additional ATP generation in tumor cells. Importantly, the folate cycle is an important source of NADPH a major cellular antioxidant synthesized in the cytoplasm as well as in mitochondria. Farber in 1949 discovered that leukemic cell growth was stimulated by folate derivative tetrahydrofolate, while antagonists, such as methotrexate, resulted in remission. Folate depletion reduces proliferation in HCC cells, suggesting folate as an important nutrient promoting HCC growth. Methylene-tetrahydrofolate dehydrogenases (MTHFD2 and MTHFD1L) are frequently upregulated in HCC, and the depletion of these mitochondrial enzymes reduces NADPH levels and causes a build-up of ROS. In addition to NADPH, the folate cycle regenerates the high-energy compound NADH, producing formate and ATP in the mitochondria of HCC cells [71,79].

## 6. Metabolism and Immune Suppressive Microenvironment

A successful immune response has certain features characterized by the synthesis of cytokines, antibodies, and cell growth. After clearing the antigen, immune cells will enter a quiescent status in which memory T cells are conserved, while the other cells experience a phase of apoptosis or senescence [80,81,82,83]. Therefore, immune activation requires a metabolic reprogramming that switches the cells from requesting basic energy and synthesis to the cells actively proliferating and secreting effector proteins. The TME which is hypoxic and acidic supports the metabolic activity of neoplastic cells dependent on glycolysis and/or glutaminolysis [24]. However, this TME causes restrain on T-effector cells (Teff), which rely primarily on glycolysis and compete with cancer cells; such metabolic competition in TME [84,85] during the immune response to an antigen triggers the T cells to interact with each other as well as with APCs forming clusters [86,87,88]. The competition between T cells and APCs (e.g., dendritic cells/macrophages) for substrates reduces these substances from the environment, and the resulting low levels of glucose induce dendritic cells to produce IL-12 and proficient cytotoxic T-cell responses to the antigen [88]. Furthermore, T-cell competition for substrates represents a selection mechanism of T cells with a high affinity for antigens that are essential for homing and migration [89,90,91]. Conversely, antigen-activated naïve T cells require glucose, glutamine, arginine, and other amino acids to differentiate into effector cytotoxic and memory T cells. Tumors with elevated rates of glycolytic activity are more capable to elude T-cell immune surveillance [92]. Indeed, a decrease in glycolytic activity by T cells can directly stimulate their apoptosis; an overexpression of the glucose transporter Glut-1 delays cell death and promotes T-cell proliferation. Since both metabolites and metabolic enzymes also act as checkpoints to regulate the behavior of T cells, insufficient levels of glucose would metabolically restrict T cells by adversely influencing the activity of, e.g., mTOR, the intensity of glycolysis, and the production of IFN-γ [93,94,95]. In a low-glucose environment, T cells use glutamine-dependent OXPHOS to produce ATP and maintain cell viability [90]. Furthermore, arginine is essential in the proper functioning and expression of T-cell receptors by regulating the translation of the ζ-chain peptide, which is required for the subsequent activation of both helper and cytotoxic T cells [96,97]. Some tumor cells increase arginine metabolism by the iNOS enzyme to assist their progression and angiogenesis, which subsequently deplete the arginine from TME [98,99]. Simultaneously, tumor-associated macrophages (TAMs), myeloid-derived suppressor cells, and dendritic cells actively consume more arginine by arginase enzyme either through increasing the activity of arginase or increasing the number of cells in the microenvironment [100,101,102,103]. Lower concentrations of arginine would lead to an inhospitable microenvironment for NK and T cells that would subsequently inhibit their anti-tumor responses [104]. HCCs are generally highly heterogeneous in their genome, metabolism, and location within the liver, which corresponds to heterogeneous TME. As a result, TME is radically different from a physiological environment in terms of composition and toxicity, and these features restrain the capability of immune cells [105,106,107,108]. Certain HCC subtypes can upregulate metabolic pathways to create a microenvironment that is highly unfavorable to anti-tumor properties of T cells [109]. This strategy is one of the mechanisms by which tumor cells prevent T cells from recognizing them and increase their immune evasion (Figure 3). The number of TAMs present in the HCC tissue correlates with an ominous prognosis for patients [109]. Macrophages are recruited to the HCC tissue from neighboring tissues and blood by the tumor itself through the secretion of chemokine/cytokine proteins [100]. Macrophages are classified into two subsets, M1 and M2; this classification is based essentially on in vitro studies, and their role in vivo is not completely unequivocal.

### 6.1. M1 and M2 Macrophages

Classically macrophages in vitro are distinct as M1 macrophages that show inflammatory functions, whereas M2 macrophages exhibit anti-inflammatory functions. Macrophage-related surface markers such as MHC class II, CD68, CD80, CD163, and CD206 are used to identify macrophage polarization. However, due the plasticity of the cells, macrophages can express pro- and anti-inflammatory surface markers simultaneously. Metabolically, M1 and M2 macrophages are defined by the expression of inducible nitric oxide synthase (iNOS) in M1 cells and by arginase-1 (Arg-1) in M2 cells [110]. M1 macrophages depend on glycolysis, while M2 macrophages rely on fatty acid oxidation and OXPHOS with glycolysis only required for the initial activation. Most TAMs suppressing the anti-tumor response and promoting tumorigenesis are generally thought to adopt a M2 polarization, but this is a restricted vision: in HCC, TAMs can express both M1 and M2 polarization markers; in the TME, TAMs trigger fatty acid oxidation to unleash immunosuppressive activity (Figure 2) [111]. Tryptophan is an amino acid playing a critical role in the proper function of both T cells and TAMs. Similar to arginine, a critical shortage in tryptophan store downregulates the ζ-Chain of the T-cell receptor [112]; tryptophan is also converted to kynurenine by the indoleamine 2,3-dioxygenase (IDO) and tryptophan-2,3-dioxygenase (TDO) enzymes of the kynurenine pathway [112]. L-kynurenine binds to a ligand-activated transcription factor, the aryl hydrocarbon receptor (Ah-R) expressed in both T cells and TAMs. The results include the generation of Treg cells and the upregulation of IDO enzyme in dendritic cells, which, in turn, suppresses the immune response to the tumor [113,114].

### 6.2. Treg Cells

Tumor expansion can be suppressed by the function of innate and acquired immunity before they appear clinically. The high infiltration of Treg cells in HCC is associated with poor clinical outcome [115]. Treg cells are a subpopulation of CD4^+^/CD8^+^ T cells playing an important role in limiting excess immune response. They determine immune tolerance to self-antigens and are also involved in feto-placental immunity [116]. HCC secreting cytokines/chemokines (e.g., CCL20, CCL2, and CSF-1) is able to recruit Treg cells and TAMs; these cells expand and remodel an HCC-favorable TME [117]. Treg cells develop from naive CD4^+^ T cells challenged by TGF-β and IL-2 in the thymus (tTreg), or peripherally derived Treg cells (pTreg) originate from conventional T cells at sites outside of the thymus. The principal Treg markers are considered CD25^+^ and FoxP3^+^ molecules; however, additional cell surface markers CD39, 5′ Nucleotidase/CD73, CTLA-4, GITR, LAG-3, LRRC32, and Neuropilin-1 contribute to defining these cells [112,113]. Noteworthy, Tregs are characterized for the expression of the FoxP3 transcription factor. FoxP3 is important for Treg differentiation and function, causing the secretion of suppressive cytokines and the expression of inhibitory surface molecules [118]. Also, the CTLA-4 molecule expressed by Treg cells prevents the maturation of APC cells and suppresses the activation of CD4^+^ T cells and CD8^+^ T cells against tumor antigens [119,120]. In HCC, the presence of regulatory CD8^+^/FoxP3^+^ T cells producing granzyme B and perforin-1 has been also described, but their role is not well defined and remains disputable. In TME, Treg cells rely on lactate and fatty acid oxidation for suppressive function, tumor cells utilize mainly glucose and glutamine; under a low-glucose condition, Teff cells fail to survive. Hence, we can infer that a reduction in glucose is detrimental for Teff-cell proliferation and cytokine production, but it is favorable for Treg cells [121,122,123,124]. Furthermore, lactate accumulation in TME causes defective growth and function of Teff cells (CD4^+^ and CD8^+^ T cells) [125,126]. Along these lines, it is conceivable that the metabolism of cancer cells and TAMs allows Treg cells to proliferate and support their suppressive function [127].

## 7. Immunoediting the HCC Tme

### 7.1. Targeting TAM Activation and Restoring Phagocytic Ability

Inhibition of macrophage recruitment could be detrimental because we are going to lose their immune stimulatory function as major phagocytes and APCs within the TME [127]. Despite generally being tumor supportive, macrophage function can be therapeutically exploited to restore their anti-tumor properties. Thus, switching TAMs toward an anti-tumor phenotype provides a more effective approach to optimizing current immunotherapies. In homeostasis, normal cells can avoid self-elimination by phagocytes through the expression of anti-phagocytosis molecules, which are, therefore, called phagocytosis checkpoints [128]. Several studies have shown that tumor cells depend on phagocytosis checkpoints to evade immune surveillance. Therefore, identification and intervention with phagocytosis checkpoints might provide a new approach for restoring the phagocytic capacity of TAMs to eliminate tumor cells [129]. Signal regulatory protein alpha (SIRPα) is an inhibitory receptor expressed on myeloid cells, including macrophages [129]. SIRPα recognizes CD47, which acts as anti-phagocytic signal and is found to be overexpressed by HCC cells and correlate with patients’ poor survival [130]. In this study, the macrophage phagocytosis of tumor cells was restored after treatment with CD47 antibodies [131], and the macrophage-mediated phagocytosis was further enhanced in the presence of chemotherapeutic drugs, suggesting that patients with lower CD47 expression were more likely to benefit from adjuvant TACE treatment. It is worth noting that CD47 is also highly expressed in cholangiocarcinoma (CCA) [132]. Interfering with the CD47-SIRPα interaction promotes phagocytosis in TAMs and consequently suppresses the progress of CCA [132]. The unique overexpression of CD47 in CCA offers an exceptional opportunity for CD47-targeted therapy. The cross-talks between innate and adaptive immune cells provide the rationale for combining phagocytosis checkpoint inhibitors with current immunotherapies. The CSF1–CSF1R axis has also been evaluated for the proliferation, differentiation, and function of macrophages. Targeting CSF1–CSF1R signaling in protumoral TAMs represents an attractive strategy to eliminate CSF1R-dependent or reprogram M2-like TAMs [133]. Independent of the mechanism of action in a substantial number of animal models, CSF1R inhibition improves T-cell responses in combination with radiation or chemotherapeutic treatments. Additionally, the CSF1–CSF1R blockade improves the efficacy of a diversity of immunotherapeutic modalities, including CD40 agonists, PD1, or cytotoxic T lymphocyte antigen 4 (CTLA-4) antagonists. The positive results of these studies have led to clinical trials combining CSF1 and/or CSF1R inhibitors with immune checkpoint inhibitors or other immunotherapies [134,135].

### 7.2. Targeting CCL20–CCR6 Axis in HCC

The chemokine axis CCL20–CCR6 represents a novel and promising target to interfere with the tumor microenvironment as CCL20 is a key contributor to the progression of HCC [136]. CCL20 acts as a chemotactic factor that attracts the lymphocytes involved in the recruitment of the pro-inflammatory IL17 producing helper T cells (Th17) and Treg cells to sites of inflammation. Indeed, HCC expression of CCL20 is associated with tumor size, tumor number, and vascular invasion, and HCC patients with high CCL20 expression have poorer overall survival in comparison with those with low CCL20 [137]. The same study has found that CCL20 produced by HCC cells recruits CCR6^+^CD5^+^ B cells and induces angiogenesis to promote tumor growth and that blocking CCL20 suppressed angiogenesis. Overall, the studies suggest that the CCL20–CCR6 axis may be a novel target for HCC treatment.

### 7.3. Targeting the “Cockpit” of HCC

Cancer cells are capable through epigenetic mechanisms to orchestrate immune cell function within the TME. This orchestration is a key factor in cancer’s ability to evade immune surveillance and resistance to therapies, including T-cell-mediated killing. Epigenetic alterations enable tumor cells to manipulate not only their own behavior but also the immune cells around them, thereby creating an environment that supports immune evasion. Histone modifications can significantly impact how tumors, including HCC, interact with the immune system by influencing the expression of immune checkpoint molecules, antigen presentation machinery, and other factors involved in immune evasion [138,139]. Histone modifications such as acetylation and methylation can influence chromatin structure, gene expression, and the tumor’s response to immune cells. These changes can either activate or repress genes involved in immune responses [140].

**Histone Acetylation**: Acetylation of histones (especially histones H3 and H4) is typically associated with gene activation. Tumors with aberrant acetylation patterns may have altered immune response gene expression. For instance, the acetylation of histones in the promoter regions of immune-related genes such as PD-L1 can lead to their overexpression, contributing to immune resistance.

**Histone Methylation**: Methylation of histones (on specific lysine or arginine residues) is often linked to gene silencing or activation, depending on the specific context [141,142,143]. H3K27me3 (histone H3 trimethylation at lysine 27) is a mark of gene silencing, and its deposition on the promoters of immune-related genes can prevent the activation of immune pathways. H3K4me3 is associated with gene activation, and its presence on the promoters of genes involved in immune recognition (such as MHC Class I) could promote immune recognition and killing by T cells. However, the loss of this modification might make tumors less detectable by T cells. H3K36me2 is a substrate of H3 lysine 36 demethylase KDM2A. In HCC, the depletion of overexpressed KDM2A coincides with the reduced presence of H3K36me2 in the promoter region of genes and leads to the re-expression of innate and adaptive immunity proteins such as PD-L1 in human HCC cells and H2-D1, H2-T23, and Mr1 in the HCC of mice.

**Chromatin Remodeling**: Modifications to the chromatin structure, such as SWI/SNF complexes, can affect the accessibility of DNA and alter the expression of immune-related genes. Tumors that undergo chromatin remodeling may either enhance or block the accessibility of immune-related genes, thus modulating the immune response [144,145].

### 7.4. Manipulating Epigenetic Mechanisms to Reprogram HCC Immunogenicity Within the TME

#### 7.4.1. Modulating Immune Cell Recruitment

Epigenetic alterations can affect the secretion of chemokines and cytokines that influence immune cell recruitment to the tumor site. Tumor cells can manipulate the immune response by epigenetically modifying their expression of these signaling molecules, directing immune cells to either support or suppress anti-tumor immunity [146,147]. Tumor cells can modify chemokine expression through epigenetic mechanisms like DNA methylation and histone modifications. For instance, tumors may alter the expression of chemokines such as CXCL5 and CCL20 to recruit immunosuppressive cells like regulatory T cells (Tregs) or macrophages, which hinder T-cell function and promote immune tolerance. Tumor cells may alter cytokine profiles through histone methylation or acetylation enabling them to promote a pro-tumorigenic inflammatory environment with the secretion of TGF-β and/or CSF-1 [148].

#### 7.4.2. Creating an Immune Favorable TME

Cancer cells use epigenetic modifications to create a TME that is conducive to immune evasion by suppressing the activation and function of immune-killing cells [149,150,151]. Epigenetic mechanisms, particularly histone acetylation and DNA methylation, can enhance the expression of immune checkpoint molecules like PD-L1 on tumor cells. This can suppress the function of T cells by binding to PD-1 on activated T cells, causing T-cell exhaustion and reducing their cytotoxic function. Tumor cells may also use these mechanisms to upregulate other immune checkpoint molecules, such as CTLA-4 or TIM-3. Epigenetic alterations in tumor cells may promote the recruitment and differentiation of Tregs within the TME [152,153]. TGF-β, for example, is often upregulated through epigenetic mechanisms to help generate these immune suppressive cells. Macrophages can accumulate in the TME, where they suppress T-cell activation and promote tumor progression. Epigenetic modifications of NF-kB, STAT3, or HIF-1α in cancer cells can enhance the production of cytokines and growth factors that favor macrophage recruitment, survival, and activation, further inhibiting immune responses [154,155,156,157].

#### 7.4.3. Preventing Immune Evasion Through Altered Antigen Presentation

Tumor cells can epigenetically regulate antigen presentation mechanisms to avoid detection and killing by T cells. By downregulating components of the MHC Class I, cancer cells reduce the ability of cytotoxic T lymphocytes to recognize and destroy tumor cells. DNA methylation of antigen presentation genes such as MHC Class I molecules and related antigen-processing genes (e.g., B2M and TAP1/TAP2) can be silenced through DNA methylation, leading to impaired antigen presentation. This allows tumor cells to escape recognition by T cells. Tumor cells may alter histone modifications at the MHC Class I promoter or the promoters of TAP1 and TAP2 genes, which are involved in antigen processing and presentation. Histone modifications can either suppress or activate the transcription of these key components, impacting the ability of T cells to recognize and kill tumor cells [158,159,160,161].

#### 7.4.4. Epigenetic Regulation of Immune Checkpoints

Epigenetics is one of the most important ways that tumor cells use to manipulate immune function through the regulation of immune checkpoint molecules (Table 2). PD-L1, CTLA-4, and other immune checkpoints can be upregulated or downregulated in response to epigenetic modifications, altering the effectiveness of T-cell-mediated responses [162].

## 8. Conclusions

The past decade has witnessed a rapid rise of HCC treatment based on immunotherapies, however, how checkpoint inhibitor-mediated immune activation remodels tumor and TME metabolism remains insufficiently understood. Cellular and tissue-based next-generation sequence technologies (e.g., single-cell RNA sequencing) are delineating the metabolic relationships of cell populations that regulate T-cell activation, infiltration, and function within TME. This new knowledge has allowed researchers to evaluate treatments aiming to modulate T-cell-suppressive signals and bypass CD8^+^ T-cell fatigue mechanisms that often develop in response to immunotherapy. The next major advances for immunotherapy will require deciphering the web/flux of information occurring between the tumor (e.g., epigenetic modifications can provide learning-induced changes concomitant with gene transcriptional changes) and TME, this information rerpresents the missing link between tumor growth and resistance during treatment. Epigenetic alterations in cancer cells are powerful tools for orchestrating immune cell function and metabolism in the TME. By modifying immune cell recruitment, immune checkpoint expression, antigen presentation, and the immunosuppressive nature of the TME, cancer cells can effectively create an environment that allows them to evade immune detection and resist immune-mediated killing. Understanding the role of epigenetics in immune modulation is crucial for developing new therapeutic strategies, such as epigenetic drugs and/or immune checkpoint inhibitors, that can reverse these immunosuppressive alterations and restore effective anti-tumor immune responses. With these aims in mind, the findings could pave the way to identify new strategies to increase the response rates and duration of treatment to overcome still-high intrinsic and acquired resistance.

## Figures and Tables

**Figure 1 biomedicines-13-01461-f001:**
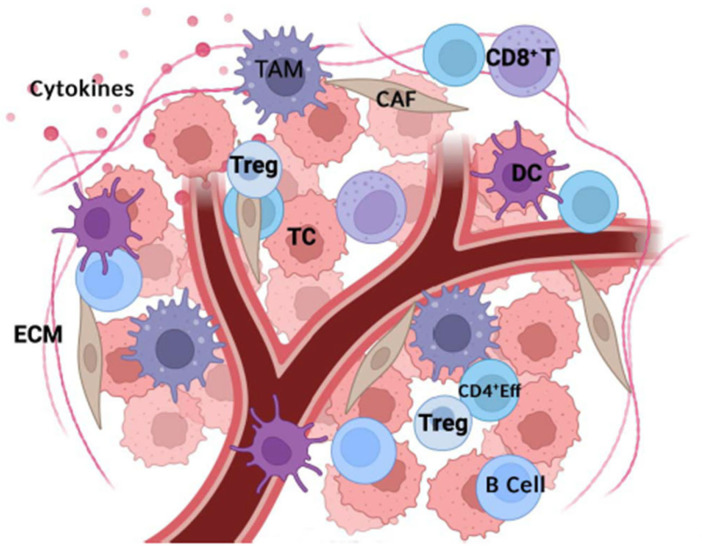
The major components of TME are blood vessels and various non-neoplastic cells such as immune cells. TME is also enriched by a variety of signaling molecules such as chemokines, cytokines, and enzymes. This cellular environment is embedded in extracellular matrix (ECM). Crosstalk between tumor cells, accessory cells, and elements of ECM all together collaborate in tumor progression and metastatic spreading. TAM, tumor-associated macrophage; DC, dendritic cell; TC, tumor cell; CAF, cancer-associated fibroblast; Treg, regulatory T cell; ECM, extracellular matrix. Created with Biorender.com

**Figure 2 biomedicines-13-01461-f002:**
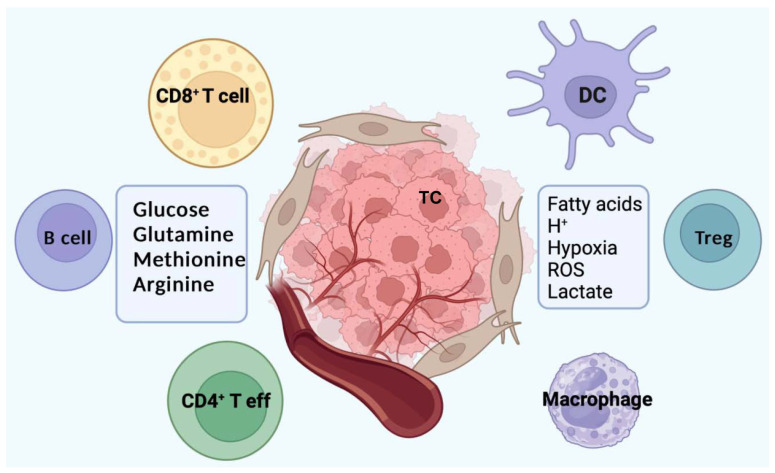
Metabolites consumed and produced by major immune cell subtypes are regrouped. Immune cell subtypes acting against HCC (TC) include CD8^+^ T cells, B cell, dendritic cell (DC), and CD4^+^ T effector. Major immunosuppressive cell types are represented by macrophages and regulatory T cells (Treg). Created with Biorender.com

**Figure 3 biomedicines-13-01461-f003:**
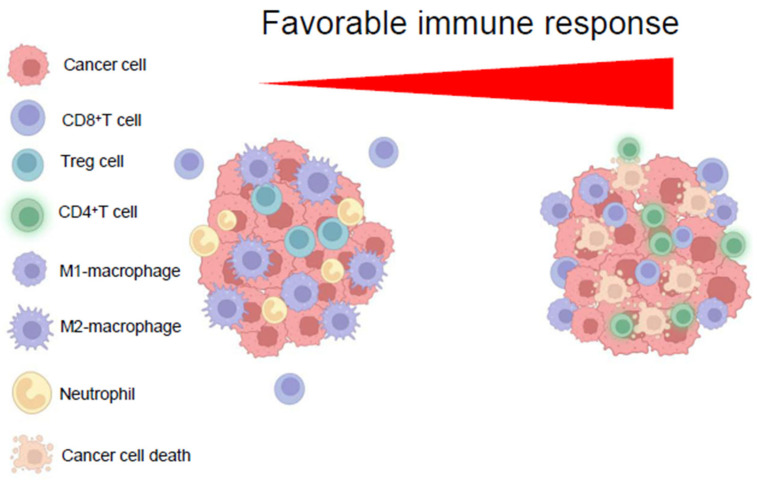
Immune cell type infiltration correlates with favorable clinical prognosis in HCC. CD8^+^ T cells have been viewed as the primary effector cells responsible for antitumor immunity; several clinical studies have also described the relevance of CD4^+^ T cells in anti-tumor immunity. A reduction in Treg cells, M2-macrophages, and neutrophils and an increase in CD8^+^/CD4^+^ ratio are the prerequisite of effective immunotherapy. Created with Biorender.com

**Table 1 biomedicines-13-01461-t001:** Immune cell metabolism and therapeutic targets.

Metabolic Pathway	Immune Cells Affected	Immune Function Effect	Therapeutic Strategies/Inhibitors
Glycolysis	- CD8^+^ T cells - Tregs - Macrophages (M1)	- Activates CD8^+^ T cells - Promotes M1 macrophages (pro-inflammatory) - Tregs may rely on glycolysis in specific contexts	- 2-Deoxyglucose (2-DG) - HIF-1a inhibitors - PI3K/Akt/mTOR inhibitors
Fatty Acid Oxidation (FAO)	- Tregs - Memory CD8^+^ T cells - Macrophages (M2)	- Enhances Treg survival (immune suppression) - Promotes M2 macrophages (anti-inflammatory) - Supports memory T cells	- CPT1A inhibitors (Etomoxir) - AMPK modulators
Glutaminolysis	- CD8^+^ T cells - Macrophages (M1) - Tregs	- Fuels effector CD8^+^ T cell proliferation - Activates M1 macrophages - May support Treg function	- Glutaminase inhibitors (CB-839) - mTORC1 inhibitors

**Table 2 biomedicines-13-01461-t002:** Interplay between epigenetics and immunotherapy.

Aspect	Cancer Epigenetics	Impact on Cancer Immunology	Therapeutic Implication
Tumor antigen expression	Epigenetic silencing of tumor-associated antigens	Reduced immunogenicity; tumors evade T-cell recognition	Epigenetic drugs can reactivate antigen expression, enhancing tumor visibility to immune system
Antigen presentation	Hypermethylation of MHC class I genes	Impaired antigen presentation to cytotoxic T cells	DNMTis can restore MHC expression and improve immune recognition
Chemokine/cytokine expression	Epigenetic suppression of immune-stimulatory chemokines (e.g., CXCL9, CXCL10)	Decreased immune cell infiltration	Epigenetic modulation can recruit T cells to the tumor microenvironment
Immune checkpoint regulation	Epigenetic upregulation of PD-L1, CTLA-4	Promotes T cell exhaustion and immune evasion	Combining epigenetic drugs + checkpoint inhibitors enhances immune response
Tumor microenvironment (TME)	Histone modifications alter stromal and immune components	Creates immunosuppressive TME	Epigenetic therapy can reprogram the TME to support anti-tumor immunity
Resistance to immunotherapy	Epigenetic plasticity enables adaptive immune escape	Limits long-term efficacy of checkpoint inhibitors	Dual therapy may overcome resistance by targeting both genetic and epigenetic escape routes

## Data Availability

No new data were created or analyzed in this study.

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
