# Peer review of "Metabolism and Immune Suppressive Response in Liver Cancer"

_biomedicines, 2025, doi:10.3390/biomedicines13061461_

Round 1
Reviewer 1 Report
Comments and Suggestions for Authors
This review examines an underexplored aspect of metabolism and immune suppressive role of macrophages and regulatory T cells, and how this contributes to restrain an anti-tumor immune response in the context of hepatocellular carcinoma (HCC). The review was overall well organized and written. Some points that should further addressed by the authors are listed as below.
- While the role of lactate, hypoxia, and nutrient deprivation in suppressing effector T cells is discussed, the molecular mechanisms linking these metabolic changes to specific immune checkpoints remain underdeveloped.
- The discussion of Tregs and macrophages could be expanded to include recent studies on metabolic crosstalk between these cells and HCC, for example, lactate-driven polarization of TAMs or Treg reliance on fatty acid oxidation.
- Including a table summarizing metabolic targets and associated therapeutic strategies would enhance practical utility.
The English could be improved to more clearly express the research.
Author Response
Comment: This review examines an underexplored aspect of metabolism and immune suppressive role of macrophages and regulatory T cells, and how this contributes to restrain an anti-tumor immune response in the context of hepatocellular carcinoma (HCC). The review was overall well organized and written. Some points that should further addressed by the authors are listed as below.
- While the role of lactate, hypoxia, and nutrient deprivation in suppressing effector T cells is discussed, the molecular mechanisms linking these metabolic changes to specific immune checkpoints remain underdeveloped.
- The discussion of Tregs and macrophages could be expanded to include recent studies on metabolic crosstalk between these cells and HCC, for example, lactate-driven polarization of TAMs or Treg reliance on fatty acid oxidation.
- Including a table summarizing metabolic targets and associated therapeutic strategies would enhance practical utility.
Response: We thank this reviewer for her/his Comments. Since this Reviewer and the other Reviewers raised many points in different issues, we believe that increasing information and references on molecular mechanisms of macrophages and Treg cells metabolism could be heavy to understand for the reader. Furthermore, this burden of new information distorts the balance of the manuscript, not exclusively focused on immune cells metabolism.
Reviewer 2 Report
Comments and Suggestions for Authors
In general, this manuscript does not serve as a strong review. The following are my concerns:
The manuscript more like a general discussion on tumor microenvironment and immune cell functions rather than specifically focusing on HCC.
The paper presents various metabolic pathways, especially in CD8 T cells. While this provides simple and useful general knowledge, it is somewhat limited. It would be improved by providing more detailed mechanisms and linking them to the HCC tumor microenvironment.
The paper lacks clinical relevance. I am more interested in understanding which therapies have specific molecular consequences or effects on immune cells, the underlying reasons for these effects, and how we can improve strategies to treat HCC patients.
HCC is a highly heterogeneous tumor type. It would be beneficial to discuss this aspect, including the role of tumor cells and immune cells, with or without therapy.
Figure 2 is not related the content in line 122-157.
Author Response
In general, this manuscript does not serve as a strong review. The following are my concerns:
Comment: The manuscript more like a general discussion on tumor microenvironment and immune cell functions rather than specifically focusing on HCC.
The paper presents various metabolic pathways, especially in CD8 T cells. While this provides simple and useful general knowledge, it is somewhat limited. It would be improved by providing more detailed mechanisms and linking them to the HCC tumor microenvironment.
The paper lacks clinical relevance. I am more interested in understanding which therapies have specific molecular consequences or effects on immune cells, the underlying reasons for these effects, and how we can improve strategies to treat HCC patients.
HCC is a highly heterogeneous tumor type. It would be beneficial to discuss this aspect, including the role of tumor cells and immune cells, with or without therapy.
Response: We thank the Reviewer for her/his comments
We agree with this Reviewer that the paper lacks clinical relevance. We added a new section (yellow highlighted) regarding innovative therapeutic strategies particularly focusing on possible Achilles’ heel of HCC
Comment: Figure 2 is not related the content in line 122-157.
Response: Amended
Reviewer 3 Report
Comments and Suggestions for Authors
Reviewer Comments
-
Please add the latest data and corresponding references to the description of tumor epidemiology in line 22.
-
The concept of "field effect" appears abruptly. Please provide further explanation to improve contextual coherence.
-
The author states "Recent data suggest that malignant transformation in HCC is not primarily caused by cancer driver mutations but epigenetic factors account for the transitions from cirrhotic tissues, dysplastic nodules and HCC." Please cite the source of this data and add appropriate references.
-
The background in the first paragraph lacks depth and overlooks the contribution of hepatitis viruses to HCC development, particularly how post-infection inflammatory responses potentially promote carcinogenic transformation of hepatocytes. Additionally, the background should mention the inconsistency in carcinogenic effects among different hepatitis virus types, such as HAV.
-
The section on tumor microenvironment (TME) composition lacks relevant references. Please provide supporting citations.
-
The critical pH value data presented in lines 84-85 lacks references. Please add appropriate citations.
-
In line 93, "In TME the oxygen concentration is significantly lower than normal healthy tissues," a comma should be added after "TME."
-
Similarly, the hypoxia-related data described in lines 93-95 requires evidence from published literature. Please add relevant references.
-
In line 117, please verify if a space is needed in "H+transport."
-
Line 122: "Glucose, fatty acids, glutamine considered" should be revised to "Glucose, fatty acids, and glutamine, considered as......"
-
The reference to "Figure 2" in line 157 appears too abruptly. Please revise to either a complete sentence or incorporate it in parentheses with the preceding text.
-
Lines 168-169: AMPK appears to directly sense changes in ATP levels rather than glucose and fructose-1,6-bisphosphate. Please verify and correct.
-
Section 6 (lines 226-320) contains extensive content with unclear logical organization. Consider adding subheadings to highlight the metabolic characteristics of different immune cells in HCC.
-
The conclusion section does not adequately summarize the entire manuscript and extends beyond the scope of liver cancer. Please rewrite to ensure the conclusion properly addresses the main themes of the article.
-
Please check for extra spaces throughout the text and ensure font consistency, as some parts of the main text appear in italics.
Author Response
We also thank this reviewer and we amended all the points raised
Comment: Please add the latest data and corresponding references to the description of tumor epidemiology in line 22.
Response: Amended
Comment: The concept of "field effect" appears abruptly. Please provide further explanation to improve contextual coherence.
Response: Amended
Comment: The author states "Recent data suggest that malignant transformation in HCC is not primarily caused by cancer driver mutations but epigenetic factors account for the transitions from cirrhotic tissues, dysplastic nodules and HCC." Please cite the source of this data and add appropriate references.
Response: Amended
Comment: The background in the first paragraph lacks depth and overlooks the contribution of hepatitis viruses to HCC development, particularly how post-infection inflammatory responses potentially promote carcinogenic transformation of hepatocytes. Additionally, the background should mention the inconsistency in carcinogenic effects among different hepatitis virus types, such as HAV.
Response: We believe that many published papers have already dedicated their interest in reviewing the role of hepatitis viruses in HCC development and we want examine an underexplored aspect of metabolism and immune suppressive role of macrophages and Treg cells.
Comment: The section on tumor microenvironment (TME) composition lacks relevant references. Please provide supporting citations.
Response: Amended
Comment: The critical pH value data presented in lines 84-85 lacks references. Please add appropriate citations.
Response: Amended
Comment: In line 93, "In TME the oxygen concentration is significantly lower than normal healthy tissues," a comma should be added after "TME."
Response: Amended
Comment: Similarly, the hypoxia-related data described in lines 93-95 requires evidence from published literature. Please add relevant references.
Response: Amended
Comment: In line 117, please verify if a space is needed in "H+transport."
Response: Amended
Comment: Line 122: "Glucose, fatty acids, glutamine considered" should be revised to "Glucose, fatty acids, and glutamine, considered as......"
Response: Amended
Comment: The reference to "Figure 2" in line 157 appears too abruptly. Please revise to either a complete sentence or incorporate it in parentheses with the preceding text.
Response: Amended
Comment: Lines 168-169: AMPK appears to directly sense changes in ATP levels rather than glucose and fructose-1,6-bisphosphate. Please verify and correct.
Response: Amended
Comment: Section 6 (lines 226-320) contains extensive content with unclear logical organization. Consider adding subheadings to highlight the metabolic characteristics of different immune cells in HCC.
Response: Amended
Comment: The conclusion section does not adequately summarize the entire manuscript and extends beyond the scope of liver cancer. Please rewrite to ensure the conclusion properly addresses the main themes of the article.
Response: Amended
Comment: Please check for extra spaces throughout the text and ensure font consistency, as some parts of the main text appear in italics.
Response: Amended
Round 2
Reviewer 2 Report
Comments and Suggestions for Authors
The authors have addressed my questions.
Author Response
.
Reviewer 3 Report
Comments and Suggestions for Authors
The authors have satisfactorily completed the requested revisions. I recommend that the manuscript be accepted for publication.
Author Response
.